# Captive Reproductive Behavior, Spawning, and Early Development of White-Barred Goby *Amblygobius phalaena* (Valenciennes, 1837) and Examined Larval Survival and Viability at Different Water Temperatures and Salinities

**Pei-Sheng Chiu** [1,2] , **Shine-Wei Ho** [1] , **Cheng-Hsuan Huang** [1] , **Yen-Chun Lee** [2] **and Yu-Hung Lin** [2,*]

[1] Mariculture Research Center, Fisheries Research Institute, Council of Agriculture, Executive Yuan, Qigu, Tainan 724028, Taiwan; pschiu@mail.tfrin.gov.tw (P.-S.C.); atramentous40@gmail.com (S.-W.H.); owenzerg@yahoo.com.tw (C.-H.H.)

[2] Department of Aquaculture, College of Agriculture, National Pingtung University of Science and Technology, Neipu, Pingtung 91201, Taiwan; lalami1218@gmail.com

\* Correspondence: yuhunglin@mail.npust.edu.tw

**Abstract:** White-barred goby *Amblygobius phalaena* is a highly valued marine ornamental fish, but its captive reproduction and early life history are poorly understood. In this study, the captive reproductive behavior, early development, and optimal temperature and salinity for the larval survival and viability of *A. phalaena* were investigated for the first time. Spawning occurred between 11:00 and 13:00, with the breeding pairs naturally spawning 24 times from 1 June 2021 to 30 June 2022. The fecundity ranged from 11,022 to 95,858 eggs per spawning event. Hatching occurred approximately 81 h and 26 min after fertilization at a temperature of 27.0 ± 0.9 °C. Newly hatched larvae had a total length (TL) of 1.91–2.03 mm with 24–26 somites. The larvae transformed into juveniles at 30 days post-hatch. Experiments were conducted at different temperatures (21, 24, 27, 30, and 33 °C) and salinities (18, 24, 30, and 36 ppt) to determine the optimal conditions for larval survival and viability. The results indicate that the most suitable conditions in terms of temperature were in the range of 21–27 °C and 30 ppt for salinity. These findings provide valuable insights for the future development of captive-breeding techniques and the commercial production of other marine ornamental gobies.

**Keywords:** marine ornamental fish; reproductive behavior; natural spawning; early development; captive breeding; larval survival

**Key Contribution:** The present study is the first report to describe the captive reproduction and early life history of white-barred goby *Amblygobius phalaena* and to establish captive-breeding techniques that could help improve the commercial production efficiency of marine ornamental goby.



## 1. Introduction

The marine ornamental fish trade is a global and continuously expanding industry [1]. From the 1980s to today, its output value has increased from USD 24–40 million per year to more than USD 300 million [2,3], and there are approximately 20–30 million marine ornamental fish on the market every year [1,4]. However, to supply market demand, the excessive harvesting of coral reef fish in the wild results in habitat destruction, threats to the survival of wild populations, and loss of biodiversity [5,6]. Therefore, reducing the reliance on wild reef fish and hunting will contribute to the sustainable development of the aquarium industry and ecosystems [1,7].

The captive breeding of coral reef fish is certainly considered an effective tool to relieve the fishing pressure on and ecosystem decay in coral reefs [1,7]. In addition, increasing the research effort on captive-breeding technology not only protects ecosystems but also promotes more captive-bred individuals to enter the aquarium market [1,7,8]. To

date, approximately 400 species of marine ornamental fish from 37 families have been reported as being successfully bred in captivity. However, only 134 captive-bred species are stably available on the market [1,9], which greatly hinders the development of the marine ornamental aquarium trade.

Gobies (Gobiidae) have great potential for the aquarium trade because of their diverse colors, peaceful behaviors, and ease of domestication and adaptation to home aquaria [10–12]. According to a previous review, gobies are an important and popular species in the marine aquarium trade, ranking top five in terms of import volume, academic research input, and commercial production number in major markets [1]. From the 1970s to today, many studies on the early development and captive breeding of marine ornamental gobies have been published, some of them focusing on behavioral observations and morphological description [13–15], while some other studies have focused on captive spawning and investigating the environmental factors of larviculture and optimal live prey feeding conditions [16,17]. However, the studies on successful breeding that have been published concentrate on a specific species (genus *Elacatinus* gobies) [11,17,18]. and there is a lack of research on other species that are also common in the aquarium market, such as genus *Amblygobius*, genus *Cryptocentrus*, genus *Fusigobius*, genus *Koumansetta*, and genus *Signigobius* gobies [9]. Because of the high diversity, feeding, and habitat complexity of marine ornamental gobies [19,20], to develop a new captive-breeding technique for marine ornamental gobies and to achieve mass production, it is, therefore, necessary to investigate the spawning, early development, and larval rearing conditions of the target species case by case, rather than directly applying the methods for other species that have been published.

White-barred goby *Amblygobius phalaena* (Valenciennes, 1837), which is distributed across the Indo-Pacific region and is native to South Africa, Australia, Japan, Thailand, Malaysia, and Indonesia [21], is a marine, reef-associated species that inhabits sand burrows, the rubble margins of algal reefs, and seagrass beds [22]. It is a commercially important aquarium fish [21], which feeds mainly on filamentous algae and harpacticoid copepods hiding in the sand [23]. Thus, in a home aquarium, *A. phalaena* can assist in keeping the tank's sandy substrate clean by filtering sand through its gills for the food it contains. The current retail price of *A. phalaena* is USD 44.99 per fish [24], which is higher than that of most genus *Elacatinus*, genus *Cryptocentrus*, genus *Koumansetta*, and genus *Signigobius* gobies. Previous studies have mostly focused on describing their reproductive biology through scuba diving in the field [25–28]. However, the knowledge of the captive reproductive behavior, spawning, and early development of *A. phalaena* is still lacking.

Moorhead and Zeng [29] indicated that there was a general lack of information on the requirements for water quality conditions for the early larvae of marine ornamental fish. Except for ammonia, nitrite, and nitrate [30], temperature and salinity significantly affect the physiological status of marine fish in the aquaculture environment [31,32]. In the early life stages of marine fish, temperature and salinity can affect their development, growth, survival, and viability independently or interactively [33,34]. In general, to determine the appropriate temperature and salinity for a specific marine ornamental larval fish, researchers have transferred fertilized eggs and newly hatched larvae into different temperature and salinity gradients and evaluated them by hatchability, larval survival, and viability [16,34,35]. However, most experiments have been conducted on pelagic spawners [34–36]. To our knowledge, many popular species in the marine aquarium trade are demersal spawners, such as clownfish, damselfish, blennies, dottybacks, and gobies [4,7,37]. Nevertheless, only a few studies on demersal spawners have been published, including on false clownfish *Amphiprion ocellaris* [38], yellowtail clownfish *Amphiprion clarkia* [39,40], and ornate goby *Istigobius ornatus* [16]. For marine ornamental fish larvae, which are sensitive to temperature and salinity, determining the most suitable conditions for larval survival by testing different levels of temperature and salinity before first feeding is the first step toward successful larviculture [34–36]. Unfortunately, this step has often been ignored in most studies on marine ornamental gobies [11,17,18]. To date, no experimental studies have assessed the influence of temperature and salinity on the larval survival of *A. phalaena*,

which undoubtedly limits the development of captive-breeding technology for this species and other marine ornamental gobies.

Thus, the present study aimed to report the first results of observations of the reproductive behavior, spawning, and early development of *A. phalaena* in captivity and to evaluate the appropriate temperature and salinity for larval survival and viability, which contributes information on the suitable early larval rearing environment.

## 2. Materials and Methods

### 2.1. Broodstock Maintenance

On 23 May 2021, 30 (*n* = 30) adult *A. phalaena* of approximately 10.4–12.1 cm in total length (TL) were obtained from a local aquarium and stored in the laboratory. Three pairs were selected as the broodstock, and both sexes were determined by the spot number on the caudal fin [26]. Each pair was maintained in three flow-through cylindrical tanks (1.53 m diameter × 0.70 m height; total volume = 600 L) at the hatchery of the Mariculture Research Center (MRC). The temperature and salinity were monitored daily with a handheld multi-parameter (XT-131M, JAQUA, Taichung, Taiwan). Other water quality parameters were monitored with a handheld meter (DO30, twinno Instruments, New Taipei City, Taiwan) and a multiparameter (HI83300, Hanna Instruments, Smithfield, RI, USA) and maintained as follows: the concentrations of dissolved oxygen, $NH_3$-N, $NO_2$-N, and $NO_3$-N were >5 mg $L^{-1}$, <0.1 mg $L^{-1}$, <1.0 mg $L^{-1}$, and <30 mg $L^{-1}$, respectively. Measurements using handheld analyzers were directly taken in each tank, and water samples from each tank were collected for nitrogenous waste analysis. The photoperiod during the preconditioning and spawning period was maintained under 12L:12D (L = light, D = dark) conditions. The broodstock were fed eel feed (Grobest Feeds Co., Ltd., Pingtung, Taiwan), which was kneaded into pellets measuring 1 cm × 1 cm. They were provided with feed once daily at a rate of approximately 2–3% of their body weight. One ceramic flowerpot was placed in each tank (13 cm diameter × 13 cm height) to provide a substrate for egg laying. Reproductive behavior and spawning frequency were visually checked daily [16].

### 2.2. Reproductive Behavior

One breeding pair (Figure 1) was observed daily from 08:00 to 09:00 and 14:00 to 15:00, and if the female was found to have a bulging abdomen, the observation time was changed to 07:00–16:00 daily to identify the spawning time more accurately. The description of reproductive behavior was based on Sunobe's method [41] of visual observation with hand drawings.

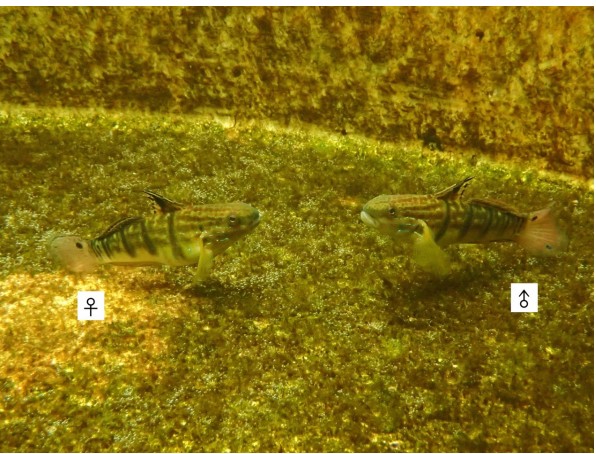

**Figure 1.** The broodstock of *Amblygobius phalaena*: male (total length, TL = 10.5 cm) is on the right, and female (TL = 11.5 cm) is on the left.

### 2.3. Spawning Record and Egg Collection

Each breeding tank was visually checked daily for new egg masses. If egg masses were found, they were separated from the substrate with a glass slide and collected by siphoning them into a small bucket, and the wet weight of each batch of eggs was measured to calculate the egg production. By counting under a microscope, it was determined that 1 g of wet weight of egg masses contained approximately 3340 eggs. After recording the number of eggs spawned, the egg masses were placed into a circular-bottom incubator (15 cm diameter $\times$ 33 cm height; total volume = 2 L). The bottom of the incubator was aerated to allow the egg masses to roll slowly into the water and to obtain sufficient dissolved oxygen. The number of eggs spawned was recorded for one year. The fertilized eggs ($n = 100$) were randomly collected and observed under a microscope (HD-2100T, FinderOptics International Ltd., Taipei, Taiwan) to calculate the fertilization rate with following the equation [42]:

$$\text{Fertilization rate } (\%) = \left( \frac{\text{The number of eggs with cleavage}}{100 \text{ eggs}} \right) \times 100$$

After calculating the fertilization rate, 100 fertilized eggs were collected and placed into an incubator until hatching. The hatching rate was calculated with the following equation [42]:

$$\text{Hatching rate } (\%) = \left( \frac{\text{Number of hatched larvae}}{\text{The number of fertilized eggs}} \right) \times 100$$

### 2.4. Embryonic Development Observation

After broodstock spawning, we transferred the egg masses into an incubator, and 30–50 fertilized eggs were collected at each sampling time for observation under a microscope (HD-2100T, FinderOptics International Ltd., Taipei, Taiwan) and photographic recording (DG-500 Calibration, FinderOptics International Ltd., Taipei, Taiwan). The embryonic development stages were recorded at 10–30 min intervals on the first day, 1–2 h on the second day, and 3 h on the third day. The morphological descriptions generally followed the terminology of Chiu et al. [16] and Kimmel et al. [43].

### 2.5. Live Prey Preparation

Different strains of cultured plankton (ciliate *Euplotes* sp., rotifer *Brachionus rotundiformis*, and copepods *Apocyclops royi*) were used as live prey for rearing *A. phalaena* larvae. *Euplotes* sp. was cultured in a 500 L fiberglass tank and fed baker's yeast. The water temperature ranged from 26.0 to 30.0 °C and the salinity ranged from 25.0 to 28.0 ppt during the entire culture period. The culture method for *Euplotes* sp. was modified from the study by Tarangkoon et al. [44]. *B. rotundiformis* were cultured in a 4000 L fiberglass tank and fed a HUFA-enriched *Chlorella vulgaris* diet (Super Fresh Chlorella-V12, Chlorella Industry Co. Ltd., Tokyo, Japan). The water temperature ranged from 28.0 to 30.0 °C and the salinity ranged from 25.0 to 30.0 ppt during the entire culture period. *Euplotes* sp. and *B. rotundiformis* were concentrated in a bucket with a plankton net (20 and 80 μm mesh, respectively) and rinsed with clean water before being fed to fish larvae. Cyclopoid copepod *A. royi* was chosen because of its wide size range and high content of essential highly unsaturated fatty acids, which are important for larval survival and growth [45]. *A. royi* nauplius were separated from copepodites and adults using 100 μm mesh. The copepods were cultured as described by Blanda et al. [46]. The size of each live prey was as follows: 60 $\times$ 80 μm for *Euplotes* sp., 70 $\times$ 100 μm for *A. royi* nauplius, 110 $\times$ 170 μm for *B. rotundiformis*, 120 $\times$ 450 μm for *A. royi* copepodites, and 140 $\times$ 600 μm for *A. royi* adults.

### 2.6. Larval Rearing

After broodstock spawning, we transferred the egg masses into incubators, where they were held until hatching. Newly hatched larvae were transferred from the incubators into a 2000 L circular fiberglass tank. The water temperature, salinity, dissolved oxygen, and

pH were maintained at 27.0 ± 0.9 °C, 30.5 ± 0.5 ppt, 6.5 ± 0.5 mg L$^{-1}$, and 8.0 ± 0.2, respectively. The photoperiod during the rearing period was maintained under 12L:12D light conditions. The feeding protocol for larval rearing was as follows: fresh *Chlorella vulgaris*-V12 (Chlorella Industry Co. Ltd., Tokyo, Japan) at a density of 10$^6$ cells mL$^{-1}$ from newly hatched (0 days post-hatch (dph)) to 30 dph; *Euplotes* sp. at 5 ciliates mL$^{-1}$ from 0 to 10 dph; *B. rotundiformis* at 5 rotifers mL$^{-1}$ from 1 to 25 dph; *A. royi* nauplius at 3 copepods mL$^{-1}$ from 0 to 30 dph; copepodite and adult *A. royi* at 5 copepods mL$^{-1}$ from 20 to 50 days post-hatch (dph); and artificial feed (eel starter feed, Grobest Feeds Co., Ltd., Pingtung, Taiwan) at 0.5 g L$^{-1}$ from 30 dph until the termination of the experimental rearing period at 55 dph (Figure 2). The densities of the different live prey were checked every day at 09:00 and 16:00 and maintained at the densities required for the experiment.

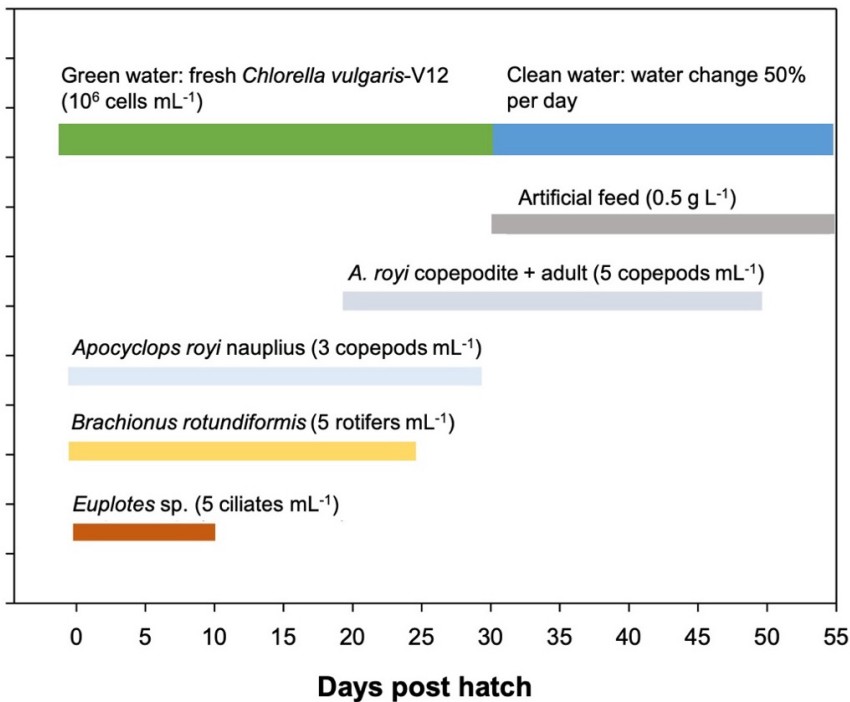

**Figure 2.** Feeding protocol for larviculture of *Amblygobius phalaena*. Prey densities at which feed was maintained.

### 2.7. Observation and Measurement of Larvae and Juveniles

The larval specimens used for observation and measurement were obtained from three breeding pairs' eggs. The larvae were sampled with a beaker and euthanized in tricaine methanesulphonate (approximately 20 mg L$^{-1}$ MS-222, Sigma-Aldrich, St Louis, MO, USA). These specimens were then observed under a microscope (SZH10, Olympus Co., Ltd., Tokyo, Japan) and with photographic recording (DG-500 Calibration, FinderOptics International Ltd., Taipei, Taiwan) and preserved in 7% buffered formalin. Morphological descriptions generally followed the terminology of Chiu et al. [16], Kimmel et al. [43], and Kendall Jr. et al. [47]. The following morphometric measurements were recorded: TL, standard length (SL), snout length (SnL), eye diameter (ED), body depth (BD), and head length (HL). Gape height (GH) was calculated using the formula [48]:

$$GH = \sqrt{(UJL^2 + LJL^2)}$$

where UJL is upper jaw length, and LJL is lower jaw length.

*2.8. Experiment 1: Effects of Different Temperatures on Larval Survival and Viability*

The experimental methods used were modified from the studies by Chiu et al. [16] and Leu et al. [34]. Before the experiment, egg masses were placed in an incubator and provided with gentle aeration until hatching. The initial water temperature was maintained at 27.0 ± 0.5 °C. When the larvae hatched, they were then transferred into different 1 L beakers containing 27.0 ± 0.5 °C seawater. Each beaker contained 40 newly hatched larvae and was provided with gentle aeration. The salinity in each treatment was maintained at 30.5 ± 0.5 ppt. Four replicate beakers for each treatment were placed in a 100 L water bath according to the group and equipped with a precision temperature-controlled heater (ADP 200 W, Mr. Aqua, Taichung, Taiwan) and a cooler (HC-130A, HAILEA, Guangdong, China). When the experiments started, temperatures were raised or lowered at a rate of 1 °C per 20 min until the set temperatures (21, 24, 27, 30, and 33 °C) were achieved. The daily survival percentage and survival activity index (SAI) [49] were calculated by counting and removing dead larvae from each beaker with a pipette.

The 2 dph larval survival was calculated using the following equation, which was modified from the study by Chiu et al. [16]:

$$\text{Survival (\%)} = \left( \frac{\text{Number of 2 dph residual larvae}}{\text{Total initialnumberoflarvae}} \right) \times 100$$

The SAI aims to assess the viability of larvae in different environmental conditions without feeding and was calculated using the following equation:

$$\text{SAI} = \frac{\sum_{i=1}^{K}(N - hi) \times i}{N}$$

where N is the total number of larvae initially present in the beaker, hi is the cumulative mortality by the ith day, and K is the number of days elapsed until all larvae died or the experiment ended.

*2.9. Experiment 2: Effects of Different Salinities on Larval Survival and Viability*

The experimental methods used were modified from the studies by Chiu et al. [16] and Leu et al. [34]. Before the experiment, the eggs were transferred from the breeding tanks into an incubator and provided with gentle aeration until hatching, as previously described. The salinity was initially maintained at 30.0 ± 0.5 ppt. When the larvae hatched, they were transferred into different 1 L beakers, each containing 30.0 ± 0.5 ppt seawater. The temperature in each treatment was maintained at 27.0 ± 0.5 °C. Each beaker contained 40 larvae, with four replicate beakers for each treatment. Salinities were raised or lowered by adding a sea salt and freshwater dilution at a rate of 2.0 ppt per h until the set salinities (18, 24, 30, and 36 ppt) were achieved. Larval survival and SAI were calculated using the equations described in Experiment 1.

*2.10. Statistical Analyses*

The relationships between annual number of eggs spawned and water quality parameters were analyzed using simple linear regressions. Biological and analytical data were expressed as means ± the standard error of the means (SEM). The survival and SAI data of Experiment 1 and Experiment 2 were compared using one-way ANOVA. Tukey's HSD test was used for multiple comparisons between means if significant overall treatment effects were found. Prior to these analyses, the survival data were normalized with an arcsine square root transformation [50]. A statistical probability of $p < 0.05$ was accepted as demonstrating a significant result in all tests. All statistics were performed using SigmaStat version 3.5 (Systat Software, San Jose, CA, USA).

## 3. Results

### 3.1. Reproductive Behavior

The results show the reproductive behavior of one breeding pair (Figure 3A). The spawning time occurred between 11:00 and 13:00. Before the spawning, the female gently pecked the male's tail with her snout every 3–5 min for approximately 20–30 min. The frequency of this pecking behavior increased as the spawning time neared (Figure 3B). Then, the male swam to the tank wall and cleaned the wall surface (Figure 3C). Afterward, the male left temporarily while the female swam to the freshly cleaned surface and laid eggs. Laying times ranged from 10 to 60 s (Figure 3D). Then, the male swam to the site of the egg masses and spread sperm. The time for releasing semen was approximately 3 s. The female's egg laying and the male's semination were repeated several times and lasted for approximately 1 h. The egg masses gradually thickened (Figure 3E). After spawning, the male fanned the water with its pectoral fins to produce enough dissolved oxygen for the egg masses. The female dwelling neared the egg clutches. The male guarded and cared for the egg masses until hatching (Figure 3F). The duration of the entire reproductive behavior was approximately 1–2 h. The male guarding periods lasted 3 or 4 days after spawning.

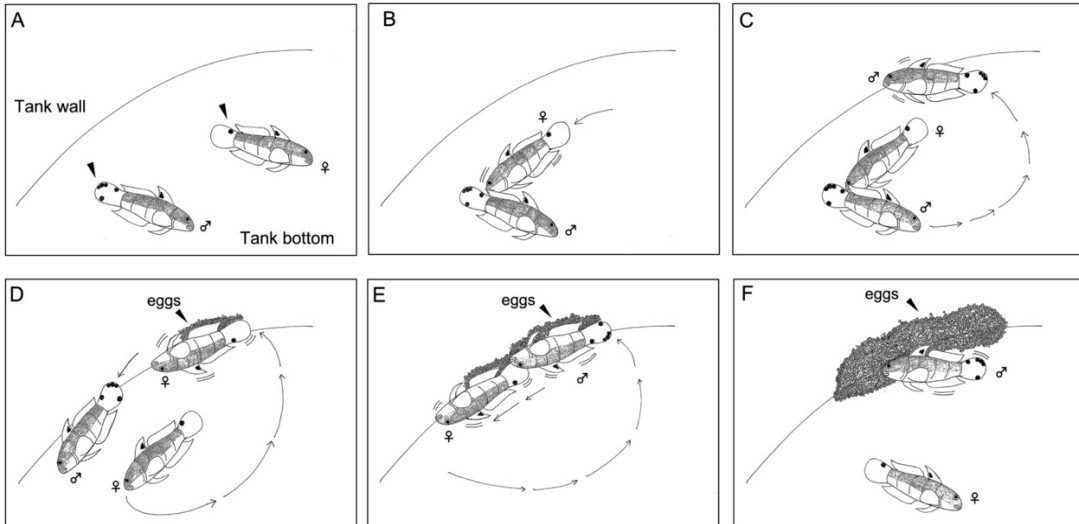

**Figure 3.** Reproductive behavior of *Amblygobius phalaena* in captivity: (**A**) the male and the female were ready to spawn on the tank wall, the male had five black spots on the caudal fin (triangle), and the female had only one black spot (triangle); (**B**) before the spawning, the female gently pecked the male's tail with her snout; (**C**) the male swam to the tank wall and started to clean the surface; (**D**) the male left temporarily while the female swam to the freshly cleaned surface and laid eggs; (**E**) the male swam to the site of the egg masses and spread sperm; and (**F**) after spawning, the male fanned the water with its pectoral fins to produce enough dissolved oxygen for the fertilized eggs, while the female dwelling neared the egg clutches. The male guarded and cared for the egg masses until hatching.

### 3.2. Spawning

The daily changes in water temperature, salinity, and the number of eggs spawned by *A. phalaena* are presented in Figure 4. Throughout the entire spawning period, the water temperature fluctuated between 20.0 and 32.6 °C (average 27.1 ± 0.2 °C), and the salinity fluctuated between 15.8 and 33.0 ppt (average 28.9 ± 0.1 ppt). The breeding pairs naturally spawned 24 times from 1 June 2021 to 30 June 2022. The average fecundity was 46,288 ± 6103 eggs, ranging from 11,022 to 95,858 eggs per spawning event, and the fertilization rate ranged from 84.0% to 97.8% with an average of 92.3 ± 2.3% ($n = 10$). The hatching rate ranged from 52.4% to 98.8% with an average of 83.3 ± 7.8% ($n = 10$) under artificial incubation conditions. There was no significant ($p > 0.05$) correlation between

water temperature, salinity, and number of eggs spawned. The spawning interval was 8–34 days (average 15.2 ± 1.5 days). Spawning occurred approximately 1–13 days (average 6.3 ± 0.8 days) before the new and full moon, including two times at the new moon and one time at the full moon.

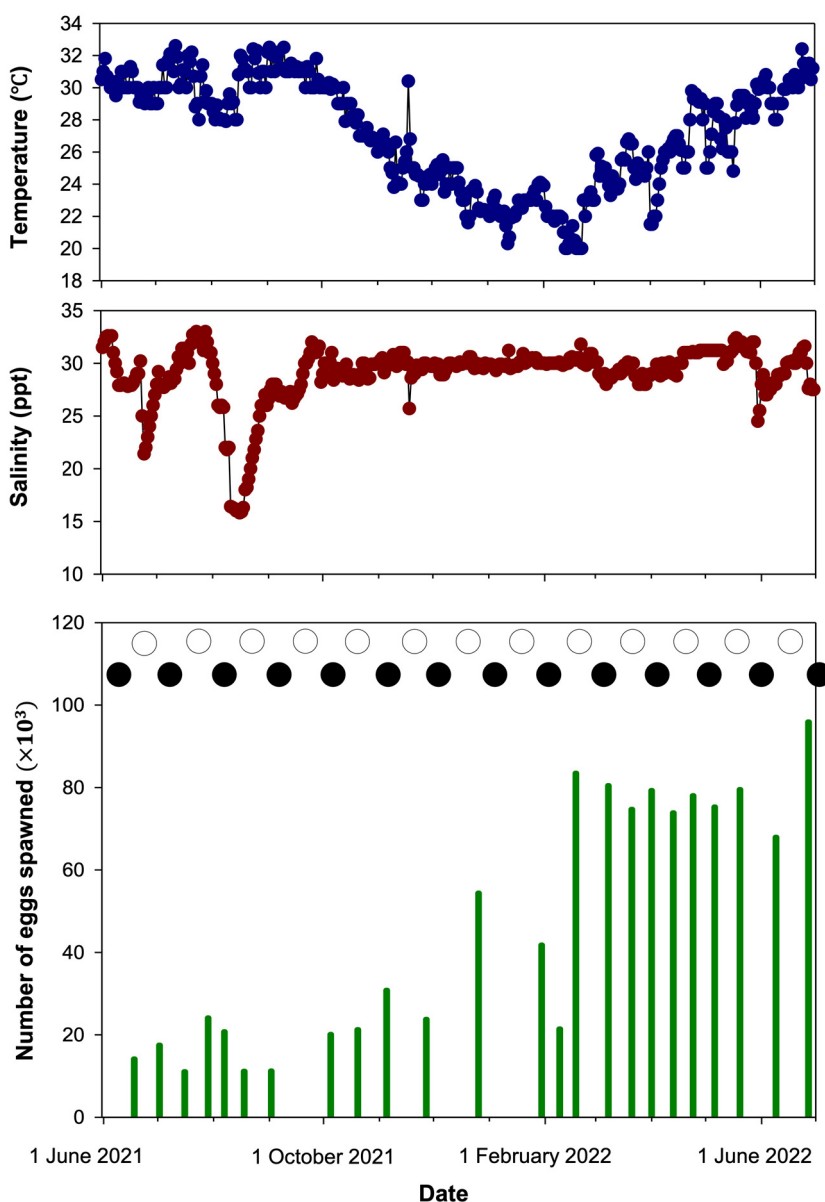

**Figure 4.** Daily changes in the water temperature (blue circles), salinity (red circles), and number of eggs spawned by *Amblygobius phalaena* during the 2021–2022 spawning seasons. Number of spawners includes one male and one female. ● = new moon; ○ = full moon.

### 3.3. Embryonic Development

The key characteristics of the embryonic development stage are summarized in Table 1. Hatching occurred 81 h 26 min (approximately 81–83 h) after fertilization at a water temperature of 27.0 ± 0.9 °C. The fertilized eggs were adhesive, demersal, and ellipsoid-shaped, with a rotund yolk and attachment filaments. The length (L) of the fertilized eggs was 1.21–1.44 mm (1.32 ± 0.02 mm, *n* = 20), while the width (W) was 0.40–0.47 mm (0.44 ± 0.01 mm; *n* = 20). At fertilization (the one-cell stage), the eggs had not yet begun to cleave, and cytoplasm streamed toward the animal pole to form the blastodisc (Figure 5A). The eggs reached the two-cell stage at 7 min post-fertilization (pf) (the first cleavage),

dividing the blastodisc into two blastomeres (Figure 5B). The 4-, 8-, 16-, 32-, and 64-cell stages were reached at 10 min pf (Figure 5C), 36 min pf (Figure 5D), 39 min pf (Figure 5E), 48 min pf (Figure 5F), and 1 h 16 min pf (Figure 5G), respectively. At 3 h 42 min pf, the high-cell stage was reached, at which the blastomeres became reduced in size (Figure 5H). At 28 h 48 min pf, epiboly covered 60% of the yolk (Figure 5I). At 31 h 46 min pf, epiboly covered 90% of the yolk (Figure 5J). At 32 h 34 min pf, the first somite appeared (Figure 5K). At 34 h 23 min pf, the six-somite stage was reached, and optic vesicles appeared. The shape of the embryo formed, and the tail bud became more prominent (Figure 5L). At 36 h 43 min pf, the 16-somite stage was reached, at which the tail bud lifted off from the yolk (Figure 5M). At 37 h 59 min pf, the 26-somite stage was reached (Figure 5N), in which the lens placode appeared. At 68 h 04 min pf, the high-pec stage was reached (Figure 5O), and the eyes' pigmentation and the fin fold appeared. The pectoral fin bud's height was approximately equal to the width of its base. At 80 h 13 min pf, the pec-fin stage was reached (Figure 5P), in which the pectoral fin now had a flat flange. The tail lengthened and looped around to reach the middle of the head. The heart was discernible. At 81 h 26 min pf, the larva hatched with a small yolk sac (Figure 5Q).

**Table 1.** Key morphological characteristics at embryonic developmental stage of *Amblygobius phalaena* cultured at 27.0 ± 0.9 °C.

| Developmental Stage | Duration Time (h:min pf) | Key Morphological Characteristic |
| --- | --- | --- |
| Zygote | 00:00 | 1-cell stage; cytoplasm streamed toward animal pole to form the blastodisc |
| Cleavage | 00:00 | 2-cell stage; 1st cleavage, dividing the blastodisc into 2 blastomeres |
|  | 00:10 | 4-cell stage; 2nd cleavage |
|  | 00:36 | 8-cell stage; 3rd cleavage |
|  | 00:39 | 16-cell stage; 4th cleavage |
|  | 00:48 | 32-cell stage; 5th cleavage |
|  | 01:16 | 64-cell stage; 6th cleavage |
| Blastula | 03:42 | High-cell stage; the blastomeres reduced in size |
| Gastrula | 28:48 | 60% epiboly completion |
|  | 31:46 | 90% epiboly completion |
| Segmentation | 32:34 | 1-somite stage; first somite furrow |
|  | 34:23 | 6-somite stage; optic vesicles appeared |
|  | 36:43 | 16-somite stage; the tail bud lifts off the yolk |
|  | 37:59 | 26-somite stage; the lens placode appeared |
| Pharyngula | 68:04 | High-pec; the pectoral fin bud's height was approximately equal to the width of its base |
| Hatching | 80:13 | Pec-fin stage; the pectoral fin now had a flat flange |
|  | 81:26 | Hatching; larva is free from the membrane |

h: hours; min: minutes; pf: post-fertilization.

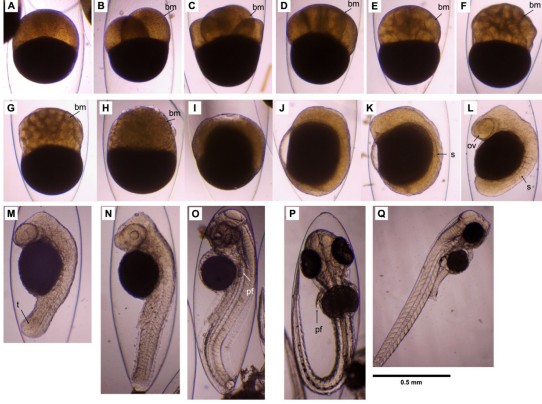

**Figure 5.** Embryonic development of *Amblygobius phalaena*: (**A**) 1-cell stage; (**B**) 2-cell stage; (**C**) 4-cell stage; (**D**) 8-cell stage; (**E**) 16-cell stage; (**F**) 32-cell stage; (**G**) 64-cell stage; (**H**) high-cell stage; (**I**) 60%-epiboly stage; (**J**) 90%-epiboly stage; (**K**) 1-somite stage; (**L**) 6-somite stage; (**M**) 16-somite stage; (**N**) 26-somite stage; (**O**) high-pec stage; (**P**) pec-fin stage; and (**Q**) egg membrane breakthrough by larva. bm: blastomere; ov: optic vesicle; pf: pectoral fin; s: somite; t: tail bud. The scale bar = 0.5 mm.

### 3.4. Larval and Juvenile Development

The key characteristics of larval and juvenile developmental stages are summarized in Table 2. Newly hatched larvae measured 1.91–2.03 mm in TL (1.96 ± 0.01 mm, *n* = 12) with 24–26 (9–10 + 15–16) somites, which were slightly curved around the small yolk sac (Figure 6A). At this stage, the gas bladder was inflated, and the eyes were fully pigmented. The mouth and anus were not open yet. The melanophores were scattered on the ventral parts of the trunk and the end of the gut. At 1 dph, the larvae were 1.94–2.05 mm in TL (1.99 ± 0.02 mm; *n* = 7) (Figure 6B). At this stage, the mouth and anus were opened, but not fully developed. The volume of the yolk sac was greatly reduced. The melanophores on the trunk and the end of the gut were denser. At 2 dph, the larvae were 2.09–2.25 mm in TL (2.15 ± 0.01 mm; *n* = 10) (Figure 6C). The yolk sac was completely absorbed. The maxilla and mandible were formed and became functional. The anus had opened. The first feeding took place at this stage, which was marked by the widening of the gut. There were 8–9 obvious melanophores scattered along the ventrolateral region of the trunk. A melanophore cluster was observed at the end of the intestine. At 10 dph, the larvae were 2.59–3.29 mm in TL (2.90 ± 0.07 mm, *n* = 12) (Figure 6D). The scattered area of melanophores on the ventrolateral region of the trunk spread out, and the number increased to 10–12. A melanophore cluster appeared at the front of the intestine. The melanophores at the end of the intestine became more obvious. At 15 dph, the larvae were 3.44–3.69 mm in TL (3.56 ± 0.02 mm; *n* = 13) (Figure 6E). The upper and lower jaws were more elongated. The area of melanophore clusters on the ventrolateral region of the trunk became more concentrated. The hypural bones formed at this stage. At 20 dph, the larvae were 5.05–6.29 mm in TL (6.06 ± 0.14 mm; *n* = 9) (Figure 6F). The hypural bones had assumed a vertical position. The second dorsal fin, anal fin, and caudal fin had appeared. At this stage, the second dorsal fin rays (II/10–11), and anal rays (I/10–11), and caudal fin rays (16–18) had developed. Numerous melanophore clusters spread from the anus along the trunk to the underside of the caudal peduncle. At 25 dph, the larvae were 8.61–9.19 mm in TL (8.79 ± 0.10 mm; *n* = 5) (Figure 6G). The first dorsal fin appeared. The pelvic fin began to develop. At 30 dph, the juveniles were 9.95–13.47 mm in TL (12.11 ± 0.48 mm; *n* = 8) (Figure 6H). At this stage, the overall adult number of fin spines and soft rays was complete, consisting of the first dorsal fin rays (V), second dorsal fin rays (II/13–15), anal rays (I/13–14), pectoral rays (10–12), pelvic rays (I/5), and caudal rays (16–18). Many melanophore clusters spread onto the head, abdomen, and base of the dorsal fins, pelvic fin, and caudal fin. At this stage, most of the juveniles began to settle down from the water column to the tank bottom or inhabited the tank wall. At 35 dph, the juveniles were 11.97–18.50 mm in TL (14.69 ± 0.84 mm; *n* = 8) (Figure 6I). There were numerous melanophores scattered on the first and second dorsal fin membranes. Four to five stripes, which consisted of melanophores, appeared on the trunk. A longitudinal black band from the snout crossed the eye and ran along the trunk to the caudal peduncle. The obvious melanophore spot appeared at the base of the caudal fin. At 41 dph, the juveniles were 19.58–25.13 mm in TL (22.07 ± 0.88 mm; *n* = 8) (Figure 6J). The body color of the juveniles changed from transparent to light green. A circular black spot appeared on the fin membrane of the first dorsal fin and the base of the caudal fin. Dark stripes appeared on the edge of the second dorsal fin and anal fin. At 52 dph, the juveniles were 37.97–53.05 mm in TL (47.24 ± 1.89 mm; *n* = 9) (Figure 6K). At this stage, the juveniles became dark brown in color, with red and black stripes on the buccal area. The first dorsal fin, second dorsal fin, and anal fin showed many bright red bands, and the pectoral fins became light yellow. Five to six dark bands on the trunk became more obvious. All juveniles settled down to the bottom and formed groups.

**Table 2.** Key morphological characteristics at larval and juvenile developmental stages of *Amblygobius phalaena* at 27.0 ± 0.9 °C.

| Developmental Stage | Duration Time (dph) | Key Morphological Characteristic |
|---|---|---|
| Larval | | |
| Pre-flexion | 0 | Larva was free from the membrane; 24–26 (9–10 + 15–16) somites; one yolk sac; gas bladder appeared; 1.96 ± 0.01 mm total length (TL; mean ± SEM) |
| | 1 | Mouth and anus opened; 1.99 ± 0.02 mm TL |
| | 2 | Yolk was completely absorbed; 8–9 obvious melanophores scattered along the ventrolateral area of trunk; 2.15 ± 0.01 mm TL |
| | 10 | The melanophores along the ventrolateral area of trunk spread and increased; 2.90 ± 0.07 mm TL |
| Flexion | 15 | The hypural bones formed; 3.56 ± 0.02 mm TL |
| Post-flexion | 20 | The hypural bones assumed a vertical position; 6.06 ± 0.14 mm TL |
| | 25 | The 1st dorsal fin appeared; 8.79 ± 0.10 mm TL |
| Juvenile | 30 | The fin ray counts attained an adult complement; 12.11 ± 0.48 mm TL |
| | 35 | A distinct melanophore spot appeared at the base of the caudal fin; 4–5 black vertical bands appeared on the trunk; 14.69 ± 0.84 mm TL |
| | 41 | The body color changed from transparent to light green; 22.07 ± 0.88 mm TL |
| | 52 | The trunk turned from light green into dark brown; several red streaks spread over the head, dorsal, and anal fin; 47.24 ± 1.89 mm TL |

dph: days post-hatch.

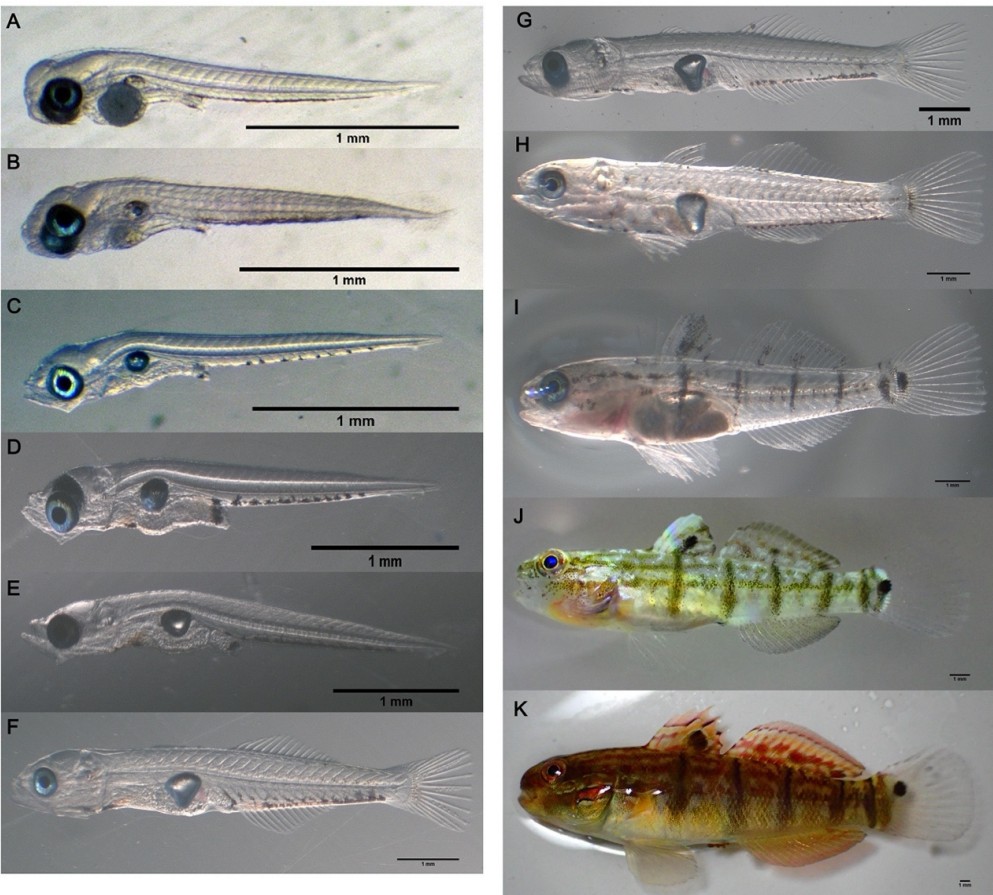

**Figure 6.** Larval development of *Amblygobius phalaena*: (**A**) newly hatched larva—1.96 ± 0.01 mm total length (TL); (**B**) 1-day post-hatch (dph)—1.99 ± 0.02 mm TL; (**C**) 2 dph—2.15 ± 0.01 mm TL; (**D**) 10 dph—2.90 ± 0.07 mm TL; (**E**) 15 dph—3.56 ± 0.02 mm TL; (**F**) 20 dph—6.06 ± 0.14 mm TL; (**G**) 25 dph—8.79 ± 0.10 mm TL; (**H**) 30 dph—12.11 ± 0.48 mm TL; (**I**) 35 dph—14.69 ± 0.84 mm TL; (**J**) 41 dph—22.07 ± 0.88 mm TL; (**K**) 52 dph—47.24 ± 1.89 mm TL. The scale bars = 1.0 mm.

At 2 dph, the mouth was opened. The gape heights were 238.45–298.21 μm (270.61 ± 9.55 μm; $n$ = 6). A quadratic equation of gape height over time was determined as follows: $Y = 502.52 − 40.69X + 2.41X^2$ (where $Y$ is the mean gape height in micrometers (μm), and $X$ represents dph). This was found to provide the best fit of the observed data and explained 99.3% ($R^2 = 0.993$; $n = 105$; $p < 0.001$) of the variation in gape size obtained during 52 dph in the culture (Figure 7).

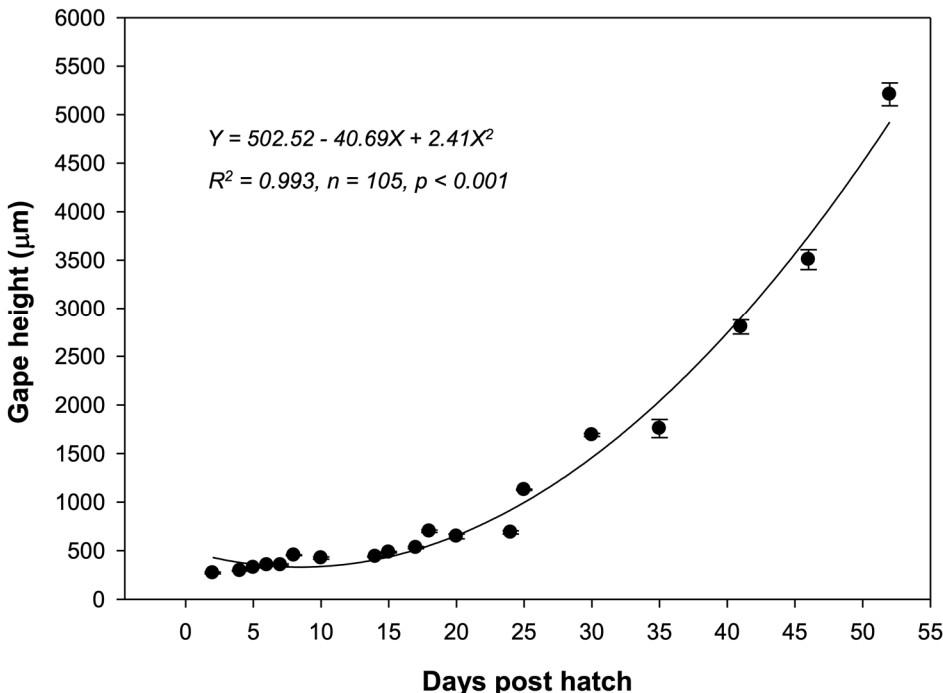

**Figure 7.** Growth in terms of gape height in reared *Amblygobius phalaena* larvae from 2 to 52 days post-hatch. All data are expressed as means ± SEM.

Quadratic equations representing the changes in each morphometric character of *A. phalaena* from hatching through the following 52 days were derived as follows: TL, $Y = 3.339 − 0.303X + 0.019X^2$; SL, $Y = 2.212 − 0.063X + 0.011X^2$; SnL, $Y = 0.1380 + 0.0061X + 0.0007X^2$; ED, $Y = 0.1900 − 0.0085X + 0.0009X^2$; BD, $Y = 0.4479 − 0.0592X + 0.0034X^2$; and HL, $Y = 0.4442 − 0.0161X + 0.0031X^2$ (where $Y$ is the mean length (mm) of the morphometric character and $X$ represents dph). These equations were determined to provide the best fit of the observed data and explained 96.8% ($R^2 = 0.986$; $n = 227$; $p < 0.001$), 98.2% ($R^2 = 0.982$; $n = 180$; $p < 0.001$), 98.5% ($R^2 = 0.985$; $n = 182$; $p < 0.001$), 99.3% ($R^2 = 0.993$; $n = 178$; $p < 0.001$), 97.7% ($R^2 = 0.977$; $n = 185$; $p < 0.001$), and 99.2% ($R^2 = 0.992$; $n = 183$; $p < 0.001$) of the variation in growth observed over the course of 52 days in the culture, respectively (Figure 8).

To complete the record of the larval development, 5 batches ($n = 5$) of larval rearing were conducted in the present study. The initial number of newly hatched larvae, harvesting time, the numbers of juveniles harvested, and the final average survival of settlement juveniles were 17,468 ± 5553 (11,022–24,048) larvae, 52–60 dph, 327 ± 87 (180–400) juveniles, and 1.92 ± 0.54 (1.56–2.84)%, respectively.

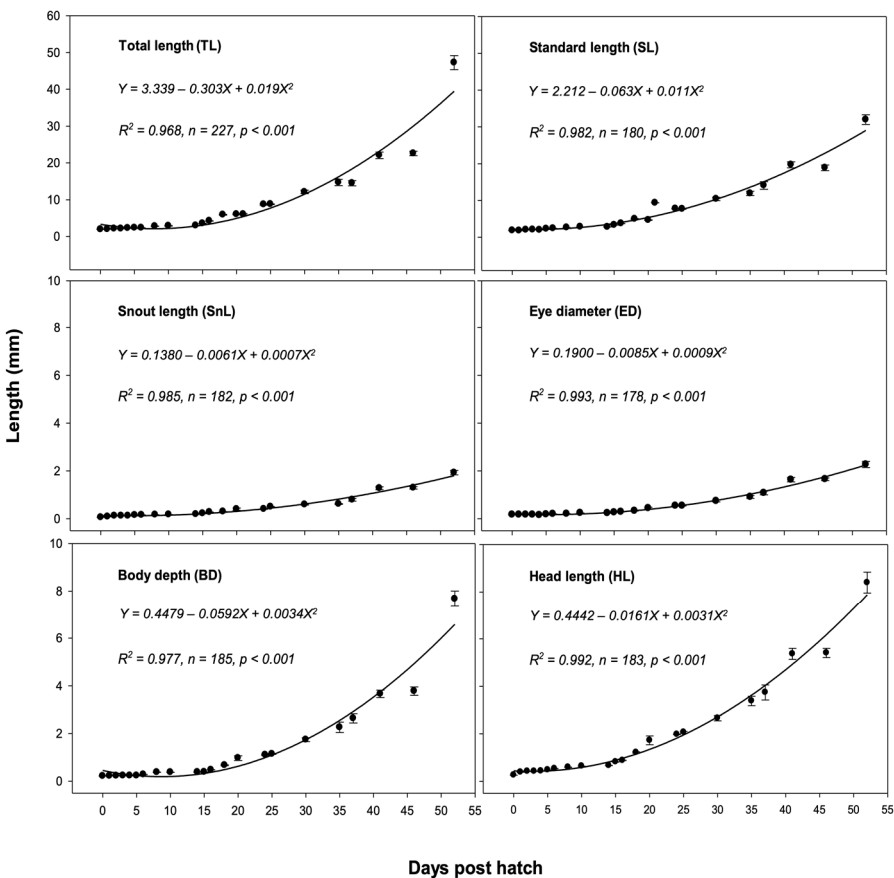

**Figure 8.** Morphometric development of total length (TL), standard length (SL), snout length (SnL), eye diameter (ED), body depth (BD), and head length (HL) of *Amblygobius phalaena* larvae over the course of 52 days in culture. All data are expressed as means ± SEM.

*3.5. Experiment 1*

The survival of 2 dph larvae was 55.6% ± 10.5%, 48.1% ± 10.1%, 50.0% ± 7.0%, 18.8% ± 3.2%, and 1.2% ± 1.2% at 21, 24, 27, 30, and 33 °C, respectively. The SAIs were 4.1 ± 0.8, 2.5 ± 0.3, 2.4 ± 0.3, 1.2 ± 0.1, and 0.6 ± 0.0, respectively. The survival in the 21 °C treatment was not significantly ($p > 0.05$) different from that in the 24 °C and 27 °C treatments, but it was significantly ($p < 0.05$) higher than that in the 30 and 33 °C treatments. The SAIs in the 21 °C, 24 °C, 27 °C, and 30 °C treatments were not significantly ($p > 0.05$) different, but the SAIs in the 21 °C and 24 °C treatments were significantly ($p < 0.05$) higher than that in the 33 °C treatment (Figure 9).

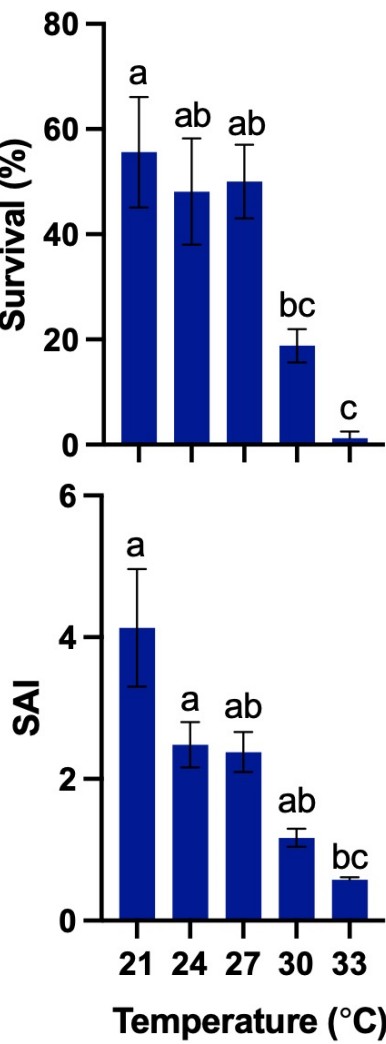

**Figure 9.** Effect of different temperatures on mean survival (%) of the 2 days post-hatch (dph) larvae and survival activity index (SAI) of *Amblygobius phalaena* larvae. All treatments were performed in four replicates. All data are expressed as means ± SEM. Means within a column followed by a different superscript letter indicate statistically significant differences from each other (ANOVA; Tukey's HSD test; $p < 0.05$).

*3.6. Experiment 2*

The survival of 2 dph larvae was 56.7% ± 4.9%, 62.5% ± 2.1%, 78.3% ± 3.9%, and 58.3% ± 5.6% at 18, 24, 30, and 36 ppt, respectively. The SAIs were 3.2 ± 0.4, 3.2 ± 0.2, 4.6 ± 0.5, and 3.6 ± 0.5, respectively. The survival in the 30 ppt treatment was not significantly ($p > 0.05$) different from that in the 24 and 36 ppt treatments, but it was significantly ($p < 0.05$) higher than that in the 18 ppt treatment. The SAIs were not significantly ($p > 0.05$) different between all treatments (Figure 10).

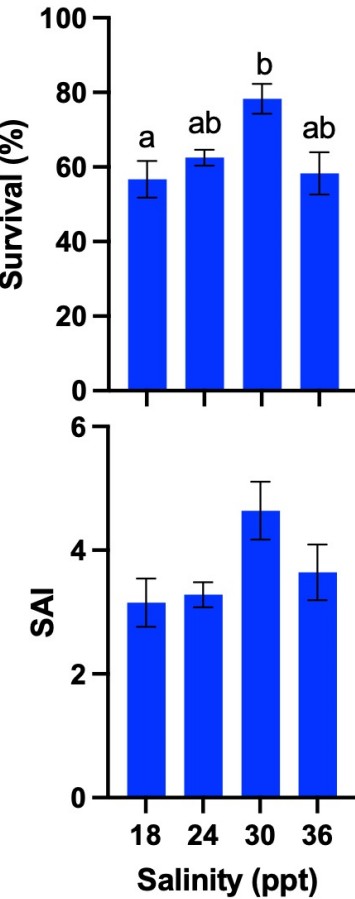

**Figure 10.** Effect of different salinities (ppt) on mean survival (%) of the 2 days post-hatch (dph) larvae and survival activity index (SAI) of *Amblygobius phalaena* larvae. All treatments were performed in four replicates. All data are expressed as means ± SEM. Means within a column followed by a different superscript letter indicate statistically significant differences from each other (ANOVA; Tukey's HSD test; $p < 0.05$).

## 4. Discussion

The present study is the first record of the reproductive behaviors of *A. phalaena* in captivity. Takegaki [26] found that *A. phalaena* usually lives in pairs and maintains territories with several burrows for shelter and spawning. However, previous studies [26] did not describe the spawning behavior and process of *A. phalaena* in detail, which was perhaps due to the limitation of diving observation vision. The findings from our study can serve as a supplement to the existing knowledge. The monogamous mating system [26] was also confirmed in the present study. We tried to put multiple females and males in the same breeding tank, but the females and males always spawned in pairs (data not published). The reproductive behavior of *A. phalaena* was triggered by the female. Similarly, the reproductive behavior of Bella goby *Valenciennea bella* also started with the female, which gently pecked the male and stimulated the male to mate [51]. In contrast, during the breeding period, it is the male shark goby *Elacatinus evelynae* that first begins to attract the female to the nest to lay eggs [11]. After spawning, male *A. phalaena* cared for and protected the egg masses, while females remain in the area near the egg masses and were not driven away by the males. Takegaki [27] indicated that if the egg-tending male was removed, the paired females would take over from the male and continue caring for the eggs until the larvae hatched. This phenomenon may be the reason why egg-tending males do not drive away paired females. The female is the candidate to take care of the egg masses and can take over from the male when he dies [27]. Reavis [52] observed that both male and female blueband goby *Valenciennea strigata* exhibited egg-caring behaviors, whereas Chiu et al. [16]

documented that the male *I. ornatus* drove the female away after spawning and guarded the eggs alone until larvae hatched. Table 3 summarizes the broodstock reproductive features of some marine ornamental gobies. Most goby species spawned on the substrates provided by the researchers, such as PVC pipes, bivalve shells, artificial caves, ceramic tiles, and gravel tiles. It is noteworthy that *A. phalaena* did not spawn in the ceramic pots that we provided; rather, they spawned on the tank walls. Herler [53] indicated that reef gobies have diverse habitats, including coral, coral-rock, and soft-sediment microhabitats, and species of *Amblygobius* prefer to inhabit the rubble near a reef's edge. In the present study, the tank wall may have been similar to a reef's edge, so the breeding pairs chose this area to spawn. Another possible reason is that the ceramic pots we provided did not fit the spawning preferences of the breeding pairs (e.g., in terms of size, material, and degree of shade). The material and size of different spawning substrates or nests should be further investigated in the future.

**Table 3.** Summary of broodstock reproductive features of marine ornamental gobies.

| Scientific Name | Common Name | Reference | Spawning Substrate | Rearing Temperature (°C) | Photoperiod (Light:Dark) | Spawning Time | Spawning Interval (Days) | Clutch Size (Eggs) |
|---|---|---|---|---|---|---|---|---|
| *Amblygobius phalaena* | White-barred goby | Present study | Tank wall | 20.0–32.6 | 12:12 | 12:00–14:00 | 8–34 | 11,022–95,858 |
| *Cryptocentrus cinctus* | Yellow prawn-goby | [15] | PVC pipes | 27–29 | – | – | 16–120 | – |
| *Elacatinus colini* | Belize sponge goby | [17] | PVC pipes | 27–28 | 14:10 | – | 6.1–9.5 | 19–388 |
| *Elacatinus figaro* | Barber goby | [54] | Bivalve shells | 26.0 | 13:11 | 7:00–10:00 | 9.1–13.3 | 430–1020 |
| | | [55] | PVC pipes and bivalve shells | 23.5–26.9 | 11:13 | 07:00–10:00 | 9.14–13.34 | 140–700 |
| *Elacatinus lori* | Linesnout goby | [17] | PVC pipes | 27–28 | 14:10 | – | – | 564–1763 |
| *Elacatinus oceanops* | Neon goby | [56] | Bivalve shells | 28 | 8:16 | – | – | 300–450 |
| *Elacatinus puncticulatus* | Panamic redhead goby | [18] | PVC pipes | 25.2–26.5 | 8:16 | 15:00–15:30 | 7–10 | 45–240 |
| *Elacatinus evelynae* | Sharknose goby | [11] | PVC pipes | 25 | 13:11 | 10:00–11:00 | – | 200–250 |
| *Eviota abax* | Sand-table dwarf goby | [57] | PVC pipes | 24–27 | – | – | – | 250–350 |
| *Eviota storthynx* | Storthynx dwarf goby | [57] | PVC pipes | 24–27 | – | – | – | 200–250 |
| *Istigobius ornatus* | Ornate goby | [16] | Artificial caves | 25.2–29.2 | 12:12 | 08:00–12:00 | 2–17 | 264–10,214 |
| *Lythrypnus dalli* | Catalina Goby | [13] | PVC pipes | 14.4–22.2 | 14:10 | 08:00–12:00 | 9.4 | 396–1055 |
| *Priolepis nocturna* | Blackbarred reef goby | [48] | Ceramic tiles | 30.0 | – | – | – | 268–3121 |
| *Trimma grammistes* | Black-striped pygmy goby | [58] | Gravel tiles | 24.5–28.0 | – | – | – | – |
| *Valenciennea bella* | Bella goby | [51] | Gravel tiles | 25.2–26.6 | – | – | 11–24 | 25,459 |

–: no data.

To our knowledge, no other publications have described year-round natural spawning data on *A. phalaena* in captivity. With year-round observation, *A. phalaena* breeding pairs spawned within 24 days. In this study, the breeding tanks used a flow-through system so the water temperature changed with the ambient environment, compared with the field environment (22–30.7 °C) [26,59], where the water temperature was slightly lower in winter and slightly higher in summer. In addition, decreases in salinity were related to heavy rainfall. Despite the greater annual variations in water temperature and salinity compared with other studies that utilized recirculating aquaculture systems [13,17,48], they did not have any significant impact on the survival and spawning of the *A. phalaena* broodstock. Furthermore, no significant correlation was observed between water temperature, salinity, and egg production. However, we observed that when the water temperature was higher than 30 °C, the broodstock still spawned as usual, but the fertilized eggs had a low hatching rate or failed to hatch. Therefore, the relationship between temperature and fertilization rate, hatching rate, and egg quality needs to be further investigated. In addition, the trend in Figure 4 shows that the number of eggs spawned was generally lower in months with high salinity variation (e.g., 1 June 2021–1 July 2021), while the number of eggs spawned gradually increased in months with a salinity range approximately 30 ppt (e.g., 1 January 2022–1 June 2022). It is possible that the salinity suitable for spawning was approximately 30 ppt, but this requires additional investigation and confirmation with statistical analysis.

Similar to our study, most studies have used a light period that is longer than or equal to the dark period for broodstock maintenance (Table 3). Only a few studies have set the light period to be shorter than the dark period for broodstock maintenance [18,55,56].

Previous studies have reported that the photoperiod significantly affects the reproduction and spawning of coral reef fishes [29,60]. More studies are needed to explore the effect of photoperiod changes on the spawning periodicity and frequency of *A. phalaena*.

The spawning of some goby species was reported to be significantly influenced by the lunar cycle [26,61,62]. In this study, the spawning of *A. phalaena* occurred before a new and full moon (approximately twice a month). Our observations were slightly different from Takegaki's [26] description of spawning 3 days before the new moon and full moon, but were generally consistent with a semilunar spawning pattern, both in captivity and in the wild. The semilunar spawning cycle was also found in starry goby *Asterropteryx semipunctata* [61], and greenbubble dwarf goby *Eviota prasine* [62]. In contrast, Kramer et al. [63] did not observe lunar spawning cycles in five species of the genus *Coryphopterus*. In captive *I. ornatus*, there was no significant relationship between egg production and the lunar cycle [16]. The information on spawning intervals can help us to estimate the spawning time of broodstock and prepare for the subsequent larviculture. The mean fecundity of captive *A. phalaena* (male: 10.5 cm TL; female: 11.5 cm TL) was 46,288 ± 6103 eggs (*n* = 24), which was slightly more than that of wild *A. phalaena* (male: 12.9–13.7 cm TL; female: 12.1–12.4 cm; 37,665–38,443 eggs; *n* = 2) [26]. On the contrary, the number of eggs spawned by *A. phalaena* is greater than that by other marine ornamental gobies (Table 3).

A wide variety of egg shapes occur in the Gobiidae, including ovoid, ellipsoid, oblong, and pear-shaped eggs, and they contain filaments for attachment to hard substrata [19]. The shape of the fertilized eggs of *A. phalaena* was similar to those of other ornamental gobies, such as *E. evelynae* [11], Catalina goby *Lythrypnus dalli* [13], and *I. ornatus* [16]. The fertilized egg size, incubation temperature, and incubation time of *A. phalaena* were compared with those of fifteen other marine ornamental goby species (Table 4). The egg size of *A. phalaena* was relatively larger than that of the Panamic redhead goby *E. puncticulatus* (0.4–0.7 mm) [18], sand-table dwarf goby *Ev. abax* (1.16–1.19 × 0.37–0.42 mm) [57], storthynx dwarf goby *Ev. storthynx* (1.19–1.31 × 0.37–0.42 mm) [57], blackbarred reef goby *Priolepis nocturna* (0.75–0.90 × 0.49–0.52 mm) [48], black-striped pygmy goby *Trimma grammistes* (0.65–0.71 × 0.45–0.50 mm) [58], *V. bella* (1.00–1.20 × 0.45–0.50 mm) [51], and *V. strigata* (1.00–1.10 × 0.29–0.30 mm) [14], whereas it was relatively smaller than those of the yellow prawn-goby *Cryptocentrus cinctus* (1.64–1.68 × 0.55 mm) [15], barber goby *E. figaro* (2.1 × 0.7 mm) [55]; (1.71–1.91 × 0.58–0.64 mm) [54], linesnout goby *E. lori* (2.49–2.59 × 0.44–0.88 mm) [17], *I. ornatus* (1.31–1.54 × 0.46–0.50 mm) [16], and *L. dalli* (1.86–1.89 × 0.46 mm) [13].

The fertilized eggs of *A. phalaena* can develop normally under temperature conditions similar to those of most tropical gobies (Table 4), whereas *L. dalli* inhabits subtropical waters and was therefore incubated at lower temperatures (18–22 °C) [13]. The incubation time of *A. phalaena* was faster those of *C. cinctus* (107 h 33 min) [15], *L. dalli* (96–247.2 h) [13], *P. nocturna* (121 h) [48], six species of genus *Elacatinus* gobies (144–169.5 h) (Table 4), and two species of genus *Eviota* gobies (110–129 h) [57], but it was longer than those of *T. grammistes* (68 h) [58] and two species of genus *Valenciennea* gobies (56 h 30 min–61 h) [14,51]. The strong inverse relationship between temperature and embryonic incubation time is well known [64]. Increasing the water temperature accelerates embryo development and reduces the hatching time but results in smaller larvae [64,65], whereas decreasing the temperature results in a longer hatching time and larger larvae [64,65]. Therefore, in future studies, understanding the suitable temperature conditions for the development of *A. phalaena* embryos may help to obtain stronger larvae and may benefit larval rearing.

**Table 4.** Summary of data on eggs, larvae, and juveniles of marine ornamental gobies.

| Scientific Name | Common Name | Reference | Egg Size (L × W, mm) | Incubation Temperature (°C) | Incubation Time | TL at Hatching (mm) | Yolk Sac Absorbed (TL/dph) | Transformation into Juvenile (TL/dph) |
|---|---|---|---|---|---|---|---|---|
| *Amblygobius phalaena* | White-barred goby | Present study | 1.21–1.44 × 0.40–0.47 | 26.1–27.9 | 81 h 26 min | 1.91–2.03 | 2.09–2.25/2 | 9.95–13.47/30 |
| *Cryptocentrus cinctus* | yellow prawn-goby | [15] | 1.64–1.68 × 0.55 | 27–29 | 107 h 33 min | 2.36–2.42 | –/– | –/– |
| *Elacatinus colini* | Belize sponge goby | [17] | – | 27–28 | 144–168 h | 3.28–3.74 [NL] | 3.5 [NL]/1 | –/38 |
| *Elacatinus figaro* | Barber goby | [55] | 2.1 × 0.7 | 25 | 168 h | 3.00 | –/3 | –/– |
| | | [54] | 1.71–1.91 × 0.58–0.64 | 26.0 | 168 h | 3.08–3.22 | –/2 | –/– |
| *Elacatinus lori* | Linesnout goby | [17] | 2.49–2.59 × 0.44–0.88 | 27–28 | 192–216 h | 3.49–3.89 [NL] | 3.5 [NL]/1 | –/28 |
| *Elacatinus oceanops* | Neon goby | [56] | 1 × 0.8 | 28 | 150–160 h | 4.00 | –/– | –/– |
| *Elacatinus puncticulatus* | Panamic redhead goby | [18] | 0.4–0.7 [D] | 25.2–26.5 | 168–169.5 h | 2.55–3.55 | –/5 | –/– |
| *Elacatinus evelynae* | Sharknose goby | [11] | – | 25 | 168 h | – | –/– | –/30–40 |
| *Eviota abax* | Sand-table dwarf goby | [57] | 1.16–1.19 × 0.37–0.42 | 24–27 | 129 h | 2.60–2.80 | 2.9–3.0/1 | –/– |
| *Eviota storthynx* | Storthynx dwarf goby | [57] | 1.19–1.31 × 0.37–0.42 | 24–27 | 110 h | 1.90–2.10 | 2.0–2.2/3 | –/– |
| *Istigobius ornatus* | Ornate goby | [16] | 1.31–1.54 × 0.46–0.50 | 27.0–28.0 | 84 h | 1.78–2.28 | 2.42–2.52/1 | 7.78–7.80/30 |
| *Lythrypnus dalli* | Catalina Goby | [13] | 1.86–1.89 × 0.46 | 14.4–22.2 | 96–247.2 h | 3.0 | –/– | –/40 |
| *Priolepis nocturna* | Blackbarred reef goby | [48] | 0.75–0.90 × 0.49–0.52 | 30.0 | 121 h | 1.85–1.93 | 1.90/1 | –/– |
| *Trimma grammistes* | Black-striped pygmy goby | [58] | 0.65–0.71 × 0.45–0.50 | 26 | 68 h | 1.80–2.20 | 2.05/2 | –/– |
| *Valenciennea bella* | Bella goby | [51] | 1.00–1.20 × 0.45–0.50 | 25.2–26.6 | 61 h | 1.65 | 1.94/2 | –/– |
| *Valenciennea strigata* | Blueband goby | [14] | 1.0–1.1 × 0.29–0.30 | 26.8–28.6 | 56 h 30 min | 0.8–1.4 | –/– | –/– |

L: length; W: width; h: hours; min: minutes; D: diameter; TL: total length; NL: notochord length; dph: days post-hatch; –: no data.

Newly hatched larvae of *A. phalaena* in this study were also compared with those of fifteen other marine ornamental goby species (Table 4). The size of the newly hatched larvae of *A. phalaena* observed during this study was smaller than those of *C. cinctus* (2.36–2.42 mm TL) [15], *E. colini* (3.28–3.74 mm notochord length) [17], *E. figaro* (3.00 mm TL [55]; 3.08–3.22 mm TL [54]), *E. lori* (3.49–3.89 mm notochord length) [17], neon goby *E. oceanops* (4.00 mm TL) [56], *E. puncticulatus* (2.55–3.55 mm TL) [18], *Ev. abax* (2.60–2.80 mm TL) [57], and *L. dalli* (3.0 mm TL) [13]. Borges et al. [19] indicated that most newly hatched larval gobies have relatively well-developed mouths, pigmented eyes, and small yolk sacs. In the present study, our observations found that the mouth of *A. phalaena* was not fully developed and functional until 2 dph, which was the time at which the yolk sac was absorbed completely. The time at which the yolk sac was completely absorbed in larval *A. phalaena* was similar to that in *T. grammistes* [58] and *V. bella* [51].

The exhaustion of the yolk sac indicated that the larval *A. phalaena* had transformed from the endogenous nutrition period to the exogenous nutrition period. In our observations, we also found that the mouths of the larvae were fully developed at 2 dph. The gape size at 2 dph of *A. phalaena* was approximately 238.45–298.21 μm, and the larvae began first feeding. Compared with other gobies, the gape size of *A. phalaena* was relatively larger than that of *L. dalli* (110 μm) [13], similar to that of *I. ornatus* (235.21–298.96 μm) [16] and *P. nocturna* (276.29–277.71 μm) [48], and smaller than that of *E. figaro* (350 μm) [54]. Hagiwara et al. [66] indicated that the mouth gape size determines the size of the prey that can be ingested. Assuming that edible prey can only be 20–50% of the maximum gape [67,68], prey must not exceed 60–150 μm to be potential food for 2 dph *A. phalaena* larvae. In this study, a mixture of *Euplotes* sp. (60 × 80 μm), *A. royi* nauplius (70 × 100 μm), and *B. rotundiformis* (110 × 170 μm) was used as the first live feed to successfully rear *A. phalaena* larvae. Therefore, all three live prey may have played an important role in the first feeding stage of *A. phalaena*. However, which live prey did the larvae ingest to survive? Or

does the ingestion of any particular live prey alone have any effect on survival? Further investigation should be conducted in the future. This will help us determine which live prey is the ideal initial prey and establish a more accurate feeding protocol.

To the best of our knowledge, information on the transformation into juvenile marine ornamental gobies is relatively scarce. Table 4 shows that only a few species were successfully raised to the juvenile stage. The reasons for the successful rearing of those species were not only the detailed descriptions of larval development in the above studies but also the investigation of the optimal water quality or suitable live prey for larval survival. Conversely, other studies without rearing to the juvenile stage are lacking regarding the optimal survival conditions of early larvae [14,57,58]. Therefore, scientists and aquaculturists still have to solve the problem of supplying a water quality suitable for larval survival and providing appropriate live prey, which may be the goal to be achieved in the future to move toward a stable commercial culture of marine ornamental gobies. In the present study, we cultivated *A. phalaena* from the hatching to the juvenile stage and harvested hundreds of completed metamorphic juveniles. Although the final survival to settlement (1.56–2.84%) was lower than in *E. evelynae* (10–50%) [11], *E. figaro* (2–30.6%) [69,70], *E. colini* (13.6–65.2%) [17] and *E. lori* (3.5–34.1%) [17], the present study is still the first record of successful larviculture of *A. phalaena*. To date, the data on the larval development of other *Amblygobius* species are limited, the larviculture techniques still need to be improved, and both issues need further investigation. The results of the present study provide useful basic information for this purpose.

To assess the effects of temperature and salinity on the larval survival of *A. phalaena*, we calculated the survival at the time when the yolk sac was completely exhausted (2 dph), which followed the method described by Chiu et al. [16] and Chiu and Leu [35]. This allowed us to exclude the effects of larval starvation. SAI values can be applied as a useful indicator to evaluate larvae viability under different environmental conditions. If larvae are moved to a specific environment, the longer they can survive without feeding the more suitable the environment or condition may be for larval survival and development [71–73]. Burt et al. [74] suggested that temperature significantly affects the early developmental stage of fish. The results of Experiment 1 show that the survival (1.2–18.8%) and SAI (0.6–1.2) of larval *A. phalaena* decreased at higher temperatures (30–33 °C) and increased (48.1–55.6% and 2.4–4.1) at relatively lower temperatures (21–27 °C). In other studies on marine ornamental fish, it has been shown that when the water temperature increases over 30 °C, there is a negative effect on the larvae, such as in *I. ornatus* (exhibiting a lower SAI at 32 °C) [16], dwarf hawkfish *Cirrhitichthys falco* (exhibiting a lower SAI and higher deformity at 30–32 °C) [35], longfin batfish *Platax teira*, and bluestreak cleaner wrasse *Labroides dimidiatus* (demonstrating a lower hatching rate and yolk sac volume at 30–33 °C) [33,34]. In studies on Melanurus wrasse *Halichoeres melanurus* and flameback pygmy angelfish *Centropyge aurantonotus*, lower survival and hatching rates were found at water temperatures of 22 °C [36], 22 °C, and 30 °C [75]. Therefore, the optimal range of incubation water temperature was species-specific. The study of the appropriate incubation temperature is an important aspect in the development of captive-breeding techniques for new species. In Experiment 1, the optimal temperature range for larval survival was 21–27 °C, although the temperature range in the habitat of *A. phalaena* was 22–30 °C [59], and the captive spawning temperature range was 20.0–32.6 °C. As mentioned before, the temperature range suitable for broodstock fish spawning may not always be suitable for larval hatching; thus, we recommend that the temperature for larval rearing should be maintained at 21–27 °C.

The results of Experiment 2 show that a salinity of 18 ppt was not suitable for the survival (56.7% ± 4.9%) of *A. phalaena* compared with a salinity of 30 ppt (78.3 ± 3.9%). Nevertheless, previous studies have pointed out that fish larval survival increases as salinity decreases [76], allowing larvae to divert more energy into growth rather than metabolism and osmoregulatory stress [77,78]. However, this may not be feasible in *A. phalaena* larvae. This may be because *A. phalaena* is a reef-dwelling goby [22,79], and it is well known that the

salinity of coral reefs is usually not less than 20 ppt. Similarly, lower salinity (< 20 ppt) has been found to reduce larval hatchability compared with higher salinity (30–35 ppt) in other reef-associated fishes, such as leopard grouper *Mycteroperca rosacea* (0–59.9% < 80.7%) [80], *L. dimidiatus* (0–28.89% < 72.22–76.67%) [34], and blue tang *Paracanthurus hepatus* (0–55% < 72.5–82.5%) [81]. In contrast, as a coral reef fish, *A. clarkii* showed higher larval survival (40–50%) and SAIs (5–6) at salinities of 15–20 ppt [39], and *I. ornatus* was broadly tolerant to salinity changes, with no significant differences in survival (68.33–88.33%) and the SAI (3.03–7.08) between salinities of 10–30 ppt [16]. This difference may be related to different fish species having different tolerances to salinity changes. Although there were no significant differences in larval survival and the SAI between 24, 30, and 36 ppt, and considering that a salinity of 30 ppt may be more suitable for *A. phalaena* broodstock spawning, we thus suggested that the appropriate salinity for larvae at this stage is 30 ppt.

Table 5 summarizes the optimal temperature and salinity conditions for larval hatching, survival, SAI, and lower deformity in some marine ornamental species. In previous studies on pelagic spawners, hatchability [34–36], larval survival [34–36], SAI [35], and deformity [34,35] were evaluated.

**Table 5.** Comparison of temperature (T) and salinity (S) conditions for optimum larval hatching, survival, SAI, lower deformity, and recommended larval rearing conditions of different species of some marine ornamental pelagic spawners and demersal spawners.

| Scientific Name | Common Name | Reference | Higher Larval Hatching | Higher Larval Survival | Higher SAI | Lower Deformity | Recommended Larval Rearing Conditions |
|---|---|---|---|---|---|---|---|
| Pelagic spawner | | | | | | | |
| *Centropyge aurantonotus* | Flameback pygmy angelfish | [75] | T: 24–28 °C | – | – | – | – |
| *Cirrhitichthys falco* | Dwarf hawkfish | [35] | T: 26–32 °C S: 27–30 ppt | T: 24–32 °C S: 24–30 ppt | T: 26 °C S: 24–30 ppt | T: 24–26 °C S: 27–33 ppt | T: 26 °C S: 27–30 ppt |
| *Halichoeres melanurus* | Melanurus wrasse | [36] | T: 28 °C | T: 25–28 °C | – | – | T: 25 °C |
| *Labroides dimidiatus* | Bluestreak cleaner wrasse | [34] | T: 26.1 °C S: 30–35 ppt | T: 27–32 °C S: 33 ppt | – | T: 22–27°C S: 30–33 ppt | T: 27 °C S: 33 ppt |
| *Paracanthurus hepatus* | Blue tang | [81] | S: 30 ppt | – | – | – | S: 30 ppt |
| *Platax teira* | Longfin batfish | [33] | T: 26.5–27.4 °C | – | – | – | T: 27 °C |
| Demersal spawner | | | | | | | |
| *Amblygobius phalaena* | White-barred goby | Present study | – | T: 21–27 °C S: 24–36 ppt | T: 21–27 °C S: 30 ppt | – | T: 21–27 °C S: 30 ppt |
| *Amphiprion clarkii* | Yellowtail clownfish | [39] | – | S: 15–20 ppt | S: 15 ppt | – | S: 15–20 ppt |
| | | [40] | – | T: 23–29 °C | – | – | – |
| *Amphiprion ocellaris* | False clownfish | [38] | T: 29–30 °C | – | – | – | T: 29 °C |
| *Istigobius ornatus* | Ornate goby | [16] | – | T: 24–32 °C S: 10–30 ppt | T: 28 °C S: 10–30 ppt | – | T: 28 °C S: 10–30 ppt |

–: no data.

However, in the case of demersal spawners, it is more difficult to sample and investigate their hatchability because of parent care. The eggs were removed and collected for experimentation using the same method as in the studies on pelagic spawners [33–35], which usually damaged the fertilized eggs. A few researchers such as Soman et al. [38] evaluated the hatchability and yolk sac volume of fertilized eggs in *A. ocellaris* by directly changing the temperature in broodstock tanks. In general, most researchers have directly investigated demersal spawner larval survival and activity after hatching [16,39,40].

Considering the results of Experiments 1 and 2, the recommended larval rearing temperature and salinity for *A. phalaena* are 21–27 °C and 30 ppt, respectively, which are similar to those of other reef fish species (25–27 °C and 27–33 ppt) [34–36]. However, some clownfish [38,39] and goby [16] larvae could tolerate higher temperatures (28–29 °C) and wider salinities (10–30 ppt). This may be one of the reasons why they were easier to culture compared with other pelagic spawners [36,81]. Most of the studies listed in Table 5 were conducted using fertilized eggs or newly hatched larvae until they entered the first feeding stage or observed the larval viability in a non-feeding state. In contrast, Dhaneesh et al. [82]

evaluated the survival of 10 dph larvae and 30 dph juvenile skunk clownfish *A. akallopisos* at different salinities, respectively. Overall, for marine ornamental fish, there are few studies that have been conducted on the optimal temperature and salinity conditions for exogenous nutritional larvae. Miller and Kendall, Jr. [83] indicated that the optimal temperature and salinity conditions for larval rearing after first feeding could be different from the optimal temperature and salinity reported for the endogenous nutritional larvae because the larvae may adapt to a completely different physiological state after feeding. Therefore, according to the experiments at this stage, we know that the optimal temperature range was 21–27 °C and salinity was 30 ppt; however, would the suitable temperature and salinity conditions for larval survival change with the live prey supply? Further research is needed.

## 5. Conclusions

This study reported the first record of the reproductive behavior and annual natural spawning of *A. phalaena* and detailed embryonic and larval development descriptions in captivity. In addition, we successfully reared *A. phalaena* from the hatching to the juvenile stage. Based on the results of Experiments 1 and 2, we recommend an optimal temperature range of 21–27 °C and salinity of 30 ppt for early larval rearing. The present study provides useful information for the future development of other marine ornamental goby captive-breeding techniques and commercial production.

**Author Contributions:** Conceptualization, P.-S.C. and Y.-H.L.; data curation, S.-W.H. and C.-H.H.; funding acquisition, P.-S.C. and Y.-H.L.; methodology, P.-S.C.; resources, P.-S.C.; supervision, Y.-H.L.; writing—original draft, P.-S.C. and Y.-H.L.; writing—review and editing, Y.-C.L. and Y.-H.L. All authors have read and agreed to the published version of the manuscript.

**Funding:** This study was financially supported by the Council of Agriculture, Executive Yuan, Taiwan under grant no. 110AS-6.5.1-AI-A2 and 111AS-6.5.1-AI-A2.

**Institutional Review Board Statement:** This study followed the protocol of the Institutional Animal Care and Use Committee (IACUC) of MRC (Permit Number 111-IACUC-02) and did not involve any endangered or protected species.

**Data Availability Statement:** All data used in this study are available from the corresponding authors upon reasonable request.

**Acknowledgments:** We thank Jinn-Rong Hseu and our colleagues for their support.

**Conflicts of Interest:** The authors declare no conflict of interest.

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
