# Peer review of "Captive Reproductive Behavior, Spawning, and Early Development of White-Barred Goby Amblygobius phalaena (Valenciennes, 1837) and Examined Larval Survival and Viability at Different Water Temperatures and Salinities"

_fishes, doi:10.3390/fishes8070364_

Round 1

Reviewer 1 Report

The topic of the manuscript is interesting and it is evident that a lot of work was done during the experiment. Let me divide my comments according to chapters of the manuscript.

Abstract - without comment

Introduction

line 56 and 58 - it is not common to cite 8 resp. 5 authors. I would recommend a max. of 3 citations per one fact (the newest ones)

line 89 and 90 and 99 -  it is not common to cite 9 resp. 7 and 5 authors. I would recommend a max. of 3 citations per one fact (the newest ones)

Materials and methods

line 113 - authors measured temperature, salinity and nitrogens - where? In each tank or in water inlet? And why didn´t you measure oxygen?

line 137 - you were calculating the eggs by weighting them - how do you know that one gramme of eggs is 3,340?

line 215 - why did you know that the best salinity for the experiment is 30.5 ppt?

line 243 - why did you know that the best temperature for the experiment is 27.0 °C?

i.e. why didn´t you make combination of this two values during the experiment? If survival rate is one of the highest at 21 °C (line 423) why won´t you change salinities also at this temperature? 

line 337 - "larval and juvenile development"

why are the "n" numbers of measured larvae changing day by day? I would understand if they went lower and lower (that larvae died) but they are increasind and decreasing (L339, new larvae, n=12; L343, 1dph, n=7; L346, 2 dph, n= 10; L351, 10 dph, n= 12 etc.). Can you explain it? Why didn´t you measure the same number of larvae?

Discussion

The discussion is well prepared, the authors discuss their results in a logical order according to the experiments. Here again, I would only recommend citing a maximum of 3 recent studies in one place. I can't specify the lines because there is no continuous number series in the discussion.

The level of the English language is very good, but there are a few ambiguities in the text and I recommend going through it again.

Only example:

line 307 - "and their width (W) was width - it can be said more clearly

line 455 - ... "can be used to supplementry"

pragraph starting "TO OUR KNOWLEDGE" - approx. in the middle - " ....... and no significant correlation between water temperature, salinity and egg production." - I miss the verb

etc.

Author Response

The topic of the manuscript is interesting and it is evident that a lot of work was done during the experiment. Let me divide my comments according to chapters of the manuscript.

Response: Thank you for your recognition of our work. We have made appropriate corrections on the revised manuscript according to your suggestions. 

Abstract - without comment

Introduction

1. line 56 and 58 - it is not common to cite 8 resp. 5 authors. I would recommend a max. of 3 citations per one fact (the newest ones)

Response: Thank you for your suggestion, we have adjusted the number of citations in the references. (L56–59).

2. line 89 and 90 and 99 -  it is not common to cite 9 resp. 7 and 5 authors. I would recommend a max. of 3 citations per one fact (the newest ones)

Response: Thank you for your suggestion, we have adjusted the number of citations in the references. (L89, 90, 99)

Materials and methods

1. line 113 - authors measured temperature, salinity and nitrogens - where? In each tank or in water inlet? And why didn´t you measure oxygen?

Response: Thank you for pointing out this issue. Regarding the water quality analysis, when using handheld analyzers, measurements are directly taken in each tank, and water samples from each tank are collected for nitrogenous waste analysis. We have now included additional information regarding dissolved oxygen measurements. (L114–121)

2. line 137 - you were calculating the eggs by weighting them - how do you know that one gramme of eggs is 3,340?

Response: First, we weighed 1 g of egg masses, and then visually counted the number of eggs in the 1 g sample under a microscope. It was determined that approximately 3340 eggs were present in the 1 g sample of egg masses. (L141–142).

3. line 215 - why did you know that the best salinity for the experiment is 30.5 ppt?

Response: Thank you for pointing out this issue. Through observation and analysis, we have inferred that the optimal spawning salinity for the broodstock is approximately 30 ppt (as mentioned in the Discussion section). As the hatched larvae would naturally be exposed to this salinity condition, we initially set the salinity to 30.5 ± 0.5 ppt and adjusted the temperature accordingly at this stage. (L215).

4. line 243 - why did you know that the best temperature for the experiment is 27.0 °C?

Response: Similarly, based on the average water temperature during the annual spawning period of the broodstock, we have set the initial temperature for this experiment as 27.0 ± 0.5 °C. This temperature condition is also closer to the observed water temperature during the embryonic and larval development. (L243).

5. i.e. why didn´t you make combination of this two values during the experiment? If survival rate is one of the highest at 21 °C (line 423) why won´t you change salinities also at this temperature?

Response: Thank you for your valuable input. At this stage, we have set the temperature and salinity as separate factors in our experiment based on the spawning conditions of the broodstock. This approach allows for easier comparison with previous studies, as many studies on coral reef fish have also utilized single-factor experiments. However, as you rightly pointed out, in our future research, we plan to design experiments using a multifactorial approach to analyze the interaction between temperature and salinity.

6. line 337 - "larval and juvenile development"

why are the "n" numbers of measured larvae changing day by day? I would understand if they went lower and lower (that larvae died) but they are increasind and decreasing (L339, new larvae, n=12; L343, 1dph, n=7; L346, 2 dph, n= 10; L351, 10 dph, n= 12 etc.). Can you explain it? Why didn´t you measure the same number of larvae?

Response: Thank you for your feedback. Our sampling procedure involved using a beaker to quickly collect larvae from the larval aggregation area in the rearing tank. The number of larvae collected in each sampling event may vary, ensuring the randomness of the sampling process and avoiding bias towards larger or smaller individuals. Inconsistent sampling numbers are common and acknowledged in previous studies (Leu et al., 2022; Chiu et al., 2023). (L343)

Discussion

1. The discussion is well prepared, the authors discuss their results in a logical order according to the experiments. Here again, I would only recommend citing a maximum of 3 recent studies in one place. I can't specify the lines because there is no continuous number series in the discussion.

Response: Thank you for your positive feedback on our manuscript. We have also made adjustments to the number of cited references accordingly.

Comments on the Quality of English Language

The level of the English language is very good, but there are a few ambiguities in the text and I recommend going through it again.

Only example:

1. line 307 - "and their width (W) was width - it can be said more clearly

Response: We have completed the revision. (L313)

2. line 455 - ... "can be used to supplementry"

Response: We have rephrased the sentence as follows: “The findings from our study can serve as a supplement to the existing knowledge.” (L461–462)

3. pragraph starting "TO OUR KNOWLEDGE" - approx. in the middle - " ....... and no significant correlation between water temperature, salinity and egg production." - I miss the verb

Response: Thank you for the comment. We have rephrased the sentence as follows: “Furthermore, no significant correlation was observed between water temperature, salinity, and egg production.”

Reviewer 2 Report

ABSTRACT

line 20: You should use a more impersonal style. Instead of " We conducted experiments with different temperatures..."  Suggestion: Experiments were conducted at different temperatures.

Line 22: Suggestion: most suitable conditions in terms of temperature were in the range of 21-27 oC and 30 ppt for salinity.

INTRODUCTION

The introduction is very well structured and provides a very good insight into the topic and addresses a larger audience.

line 105: aimed to (1) report..., and (2) evaluated. I suggest using: to report and to evaluate.

MATERIALS AND METHODS

line 113: When first mention in the text, please also add the full name of the facility (MRC).

line 119: if possible please add the size of the food structure used

line 247: Experiment 1

line 261: occurred 

line 363: The first dorsal fin appeared.

line 455: "The results of our study can be used to supplementary". Please rephrase this sentence. 

page 20 line 38:  replace "which" with a comma

page 24 line 89: citation 21 was marked with red

For aesthetic purposes, I have formatted and attached some of the used formulas for the authors.

The English language is fine. 

Author Response

Thank you for your valuable comments for our manuscript. We have made some corrections on the revised manuscript. Please refer them as below:

1. line 20: You should use a more impersonal style. Instead of " We conducted experiments with different temperatures..."  Suggestion: Experiments were conducted at different temperatures.

Response: Thank you for your suggestion, we have revised it. (L13–15, L20–21)

2. Line 22: Suggestion: most suitable conditions in terms of temperature were in the range of 21-27 oC and 30 ppt for salinity.

Response: Thank you for your suggestion, we have revised it. (L22–23)

INTRODUCTION

1. The introduction is very well structured and provides a very good insight into the topic and addresses a larger audience.

Response: Thank you for your positive comments on our paper.

2. line 105: aimed to (1) report..., and (2) evaluated. I suggest using: to report and to evaluate.

Response: Thank you for your suggestion, we have revised it. (L103–105).

MATERIALS AND METHODS

1. line 113: When first mention in the text, please also add the full name of the facility (MRC).

Response: We have added the full name of the facility. (L113–114)

2. line 119: if possible please add the size of the food structure used

Response: We have added information on food sizes. (L122–124)

3. line 247: Experiment 1

Response: We have revised it. (L253)

4. line 261: occurred

Response: We have revised it. (L267)

5. line 363: The first dorsal fin appeared.

Response: The sentence has been revised. (L369)

6. line 455: "The results of our study can be used to supplementary". Please rephrase this sentence.

Response: We have rephrased the sentence as follows: “The findings from our study can serve as a supplement to the existing knowledge.” (L461–462)

7. page 20 line 38:  replace "which" with a comma

Response: Done. (L38–39)

8. page 24 line 89: citation 21 was marked with red

Response: The color of citation has been corrected. (L88)

9. For aesthetic purposes, I have formatted and attached some of the used formulas for the authors.

Response: Thank you for your kind help, we have completed the revision. (L150, L155, L232, L237).